# Comparison of Sugar Profile between Leaves and Fruits of Blueberry and Strawberry Cultivars Grown in Organic and Integrated Production System

**DOI:** 10.3390/plants8070205

**Published:** 2019-07-04

**Authors:** Milica Fotirić Akšić, Tomislav Tosti, Milica Sredojević, Jasminka Milivojević, Mekjell Meland, Maja Natić

**Affiliations:** 1Faculty of Agriculture, University of Belgrade, 11080 Belgrade, Serbia; 2Faculty of Chemistry, University of Belgrade, 11158 Belgrade, Serbia; 3Innovation Center, Faculty of Chemistry, University of Belgrade, 11158 Belgrade, Serbia; 4Norwegian Institute of Bioeconomy Research-NIBIO Ullensvang, 5781 Lofthus, Norway

**Keywords:** *Fragaria* × *ananassa*, *Vaccinium corymbosum*, carbohydrates, HPAEC-PAD, fructose, galactose, principal component analysis

## Abstract

The objective of this study was to determine and compare the sugar profile, distribution in fruits and leaves and sink-source relationship in three strawberry (‘Favette’, ‘Alba’ and ‘Clery’) and three blueberry cultivars (‘Bluecrop’, ‘Duke’ and ‘Nui’) grown in organic (OP) and integrated production systems (IP). Sugar analysis was done using high-performance anion-exchange chromatography (HPAEC) with pulsed amperometric detection (PAD). The results showed that monosaccharide glucose and fructose and disaccharide sucrose were the most important sugars in strawberry, while monosaccharide glucose, fructose, and galactose were the most important in blueberry. Source-sink relationship was different in strawberry compared to blueberry, having a much higher quantity of sugars in its fruits in relation to leaves. According to principal component analysis (PCA), galactose, arabinose, and melibiose were the most important sugars in separating the fruits of strawberries from blueberries, while panose, ribose, stachyose, galactose, maltose, rhamnose, and raffinose were the most important sugar component in leaves recognition. Galactitol, melibiose, and gentiobiose were the key sugars that split out strawberry fruits and leaves, while galactose, maltotriose, raffinose, fructose, and glucose divided blueberry fruits and leaves in two groups. PCA was difficult to distinguish between OP and IP, because the stress-specific responses of the studied plants were highly variable due to the different sensitivity levels and defense strategies of each cultivar, which directly affected the sugar distribution. Due to its high content of sugars, especially fructose, the strawberry cultivar ‘Clery’ and the blueberry cultivars ‘Bluecrop’ and ‘Nui’ could be singled out in this study as being the most suitable cultivars for OP.

## 1. Introduction

Sugars are primary products of photosynthesis, and it was previously thought that sugars were a fundamental compound correlated with fruit quality and flavor, which determine the caloric value of the fruit. Now, it has been proved that sugars are necessary for building up elements in the cell walls and energy sources in plants, which are used as precursors for aroma compounds and signaling molecules. They are involved in nearly all fundamental processes within plant metabolism, including cell-to-cell communication, embryogenesis, seed germination and progress in seedling growth, development of vegetative and reproductive organs, senescence, responses to all kinds of biotic and abiotic stresses, coordinating the expression of many genes, and synthesis of organic and amino acids, polyphenols, pigments and aroma compounds [1,2]. Stage of ripeness, age of plants, soil quality, fertilization, region and weather conditions, cultivation, geographical origin, and genotype are known to affect the quantitative variations in sugar [3].

In order to have high quality and high yields in modern orchards, it is necessary to understand the relationship between source organs (typically mature leaves which act as the principal source of the plant of translocated sugars) and sink organs (immature leaves, fruits, seeds), which are both under the strong influence of genotype and environment [4]. It is important to underline that not all leaves produce the same amount of carbohydrates, nor do all fruits receive the same amount of sugars. Crop load and deficiency or imbalance of mineral nutrients can affect assimilate distribution and source/sink organ functioning [5].

Due to their freshly consummated highly colored fruits, delicious taste, low calories and richness of their bioactive compounds, berry fruits have been constantly increasing in popularity over the last few decades. Berry fruits have very low amounts of lipids, but are rich in dietary fibers, and are extraordinary sources of sugars, acids, phenolics, anthocyanins, carotenoids, flavonoids, tannins, vitamins (A, B1, B2, C and PP) and minerals [6]. Thus, berry-based foods are rich in antioxidants, which are effective for eliminating free radicals and reactive oxygen species that are, in most cases, the cause of chronic diseases. That is the reason nutraceuticals and functional foods have become very fashionable for people who want to ensure maximum health benefits from food [7].

Cultivated strawberry (*Fragaria* × *ananassa* Duch.), a member of the *Rosaceae* family, is one the earliest fruit species in temperate regions. It is very popular among producers due to the quick financial return and among consumers due to its delicious fruits, thus receiving a lot of attention [8]. World production of strawberry is 9,223,815 t, where China is a leading country, with ~3.7 mil t, as well as the USA, with ~1.5 mil t [9]. It is considered to be an excellent source of compounds, with high antioxidant capacity, high biological activities and potential health benefits such as reducing myocardium ischemia, reduction of thrombosis risk and anti-cancer activity [10]. Additionally, leaf extract of strawberries can be used to treat *diabetes* mellitus, against inflammation and against apoptosis (self-destructive cellular process in tissue development) [11].

Northern highbush blueberry (*Vaccinium corymbosum* L.) is one of the top-ranking berry fruit species, being very popular in Europe in recent decades [12]. Total world production of blueberry is 596,813 t, with the highest production being in the USA (236,621 t) and Canada (160,246 t) [9]. The majority of blueberries are conventionally grown, but organic cultivation has also gained popularity in recent years. Blueberries are reported to be good for ophthalmologic disorders, against osteoporosis, exhibiting anti-diabetic properties, decreasing blood pressure and blood cholesterol, and inhibiting the development of cancer cells in the breast and colon [13,14].

Recently, organic (fruit) production has become very popular due to the ‘sustainable intensification’ of the production with less pesticide and heavy metal residues, pronounced aroma, better soil health and higher biodiversity [15]. The total area under organic temperate fruit production in the world is over 204,000 ha, representing 1.6% of the whole area of temperate fruit grown worldwide. The top-ranking countries with the largest area of temperate fruit production are China, Poland, Germany, Italy, USA, Turkey and France, where the key temperate fruits are apples, with 40%, followed by pears, cherries, and plums, but with no data for berries [16].

So far, many scientific studies have been carried out comparing integrated/conventional and organic strawberry [17,18,19] and blueberry production [20,21,22,23]. In this way You et al. [24] showed that organically grown berries have significantly higher levels of bioactive compounds compared to berries from integrated production. Wang et al. [25] and Olsson et al. [26] found that blueberries and strawberries from organic production stored much higher levels of polyphenols in fruits in contrast to integrated production. In addition to fruit quality, Andersson et al. [27] showed that bees have a preference for organic strawberries, which are then better pollinated and have a lower number of deformed fruits. Reganold et al. [28] documented that higher quality of organic strawberries was associated with higher soil quality, because soil under organic production enhanced water retention, improved soil structure, fixed nitrogen, stored higher levels of carbon and macro and micro nutrients, and exhibited higher activity of soil micro-fauna. On the other hand, some studies proved no consistent differences between those two growing systems when studying the fruit of several raspberry cultivars [29].

In addition to aroma, fruit color and firmness, sugar content is one of the main parameters considered that affect fruit taste and one of the essential criterion considered in the evaluation of the nutritive value of berries, and nutritional fruit quality. Although sugar level is affected by genetic factors [30], it can be under the strong influence of the cultivation techniques and different pre-harvest conditions [31,32]. Taking into account the significant increase in consumer interest in berry fruits and organic production popularization, it is of vital importance to test the fruits and leaves of different strawberry and blueberry cultivars in two different cultivation systems in order to obtain viable information about its sugar profile. From a scientific standpoint, the objective is to compare sugar distribution and source-sink relationship in two berry fruit species under OP and IP and determine whether there are some differences in sugar metabolism between these two berry species. The knowledge obtained from this study could help us shape the source-sink relationship, which secures a high quality of fruits and economically cost-effective yields.

## 2. Results and Discussion

### 2.1. Differences in Strawberry and Blueberry Carbohydrate Profiles

In accordance with the literature data, glucose, fructose and sucrose were the most abundant sugars detected in strawberry (Table 1) [33,34].

Strawberry is a non-climacteric fruit, and has to be picked at full maturity, because afterwards, no nutritional and sensory qualities (aroma, texture and flavor) are able to develop [35]. In the case of blueberry, the most abundant sugars were glucose, fructose, and galactose (Table 2), which is in line with previously published manuscripts [36].

Characterization of carbohydrates in fruits revealed a total of 17 compounds (7 monosaccharides, 5 disaccharides, 4 trisaccharides, and 1 sugar alcohol) in strawberry samples and 15 compounds (6 monosaccharides, 4 disaccharides, 4 trisaccharides, and 1 sugar alcohol) in blueberries (Table 1 and Table 2). The contents of seven sugars (fructose, sucrose, trehalose, isomaltortiose, maltose, panose, and rhamnose) were higher in strawberries than in blueberries. In particular, it should be pointed out that sucrose was one of the major sugars in strawberries (in range from 15.863 mg/g to 68.120 mg/g), while in blueberries, its concentration did not exceed 0.149 mg/g. This is probably due to the fact that sucrose has a critical role as a signal in the maturation of strawberry, which is a non-climacteric fruit [37].

Regarding leaves, a total of 19 carbohydrates (7 monosaccharides, 6 disaccharides, 4 trisaccharides, and 2 sugar alcohols) were found in strawberry leaf samples (Table 3), while in blueberry leaves (Table 4), in addition to from those, one tetrasaccharide was quantified (a total of 20). In all leaf samples, the most abundant sugar was glucose (3.696–8.047 mg/g), followed by fructose (2.242–7.065 mg/g). Among disaccharides, sucrose was found in the highest concentrations (0.362–2.374 mg/g). In strawberry leaves and fruits, glucose + fructose + sucrose combined together accounted for 77.88–87.98% and 96.53–98.41% of all sugars detected, respectively. In case of blueberry leaves and fruits, the most abundant sugars were glucose, fructose, and galactose, the sum of which was in the range of 77.56% to 81.28% and between 99.02 and 99.36% of all sugars detected, respectively.

In regard to glucose-to-fructose ratio (G/F), leaves had a higher ratio (in favor of glucose) than fruits, but no clear line could be drawn between strawberry/blueberry and organic/integrated production. It is well known that during fruit ripening, sucrose breaks down to glucose and fructose, the amounts of which should be the same. This happened in this study too, where strawberry and blueberry fruits exhibited this ratio ~1 (except for integrated ‘Clery’). Regarding strawberry leaves, this ratio was increased to 1.65 (‘Alba’ organic production), while the ratio in blueberry was up to 1.57 (‘Nui’ integrated production). This can be explained by the fact that glucose is an essential sugar in plant metabolism; not just for fruit ripening, but also for some other structural, nuclear and biochemical processes in plants (signaling, growth, development and respiration) [38].

Turanose, isomaltose, and xylose were found only in strawberry fruit samples, while melibiose was detected only in blueberries, and could potentially be used as a marker for blueberry products. As for the leaves, rhamnose and raffinose were higher in strawberry than in blueberry leaves. Stachyose was detected only in blueberry leaves, while contents of panose were higher by up to 20 times in these samples.

### 2.2. Source to Sink Relationship between Leaves and Berries

Manipulating the source-sink relationship in order to achieve an acceptable crop load is sophisticated job due to the complicated sugar metabolism in fruits. Mostly sucrose (and sorbitol in woody *Rosaceae* species) and some small quantities of amino acids, proteins, hormones, inorganic ions are translocated in phloem as a result of the difference in osmotic pressure between source and sink organs [39].

In this experiment, chemical analysis of leaves and fruits from organic and integrated production revealed that fruits from both fruit species, as expected, stored much higher levels of main sugars. In strawberry, fruits stored up to ~12-fold higher levels of glucose, ~22-fold higher levels of fructose, and ~66-fold higher levels of sucrose compared to leaves. In blueberries, fruits collected up to ~6.5-fold higher levels of glucose, up to ~9-fold higher levels of fructose and up to ~12-fold higher levels of galactose compared to leaves. However, in blueberries, leaves accumulated up to ~23.7-fold higher levels of sucrose compared to fruits. The results of this study are going in a line with those from Petridis at al. [40], who proved that ‘Duke’ fruits stored ~4-fold higher levels of glucose and ~10-fold higher levels of fructose compared to leaf, but leaf stored ~3.5-fold higher levels of sucrose than fruit.

Translocation of sugars from leaves (source organs) to xylem cells (sieve tube) by the concentration gradient thought the plasmodesmata, and then its further transport to fruits (sink organs) is under the strong control of ecological factors (humidity, wind, drought, CO_2_ concentration, light, temperature) and all kinds of stresses [41]. According to Sun et al. [42], leaves with increased sugar content were found in strawberry plants under drought stress. OP is in most cases considered a more stressful production system, due to the insufficient supply of mineral nitrogen, the limited number of crop protection products allowed, or the inefficient application of pesticides, which leads to a higher accumulation of primary and secondary metabolic products [43]. However, our study showed that synthesis of different sugars and their ratio between fruits and leaves under divergent production systems in strawberries and blueberries was cultivar-dependent. This was in accordance with Bordonaba and Terry [33], whose results indicated that strawberry sweetness increased when grown under irrigation deficit, admitting that this consequence was mostly genotype-dependent.

### 2.3. Sugar Profiles of Plants from Organic and Integrated Production

When comparing organic and integrated production, on average, strawberry leaves and blueberry fruits from organic production stored higher levels of the three most important sugars (glucose, fructose and sucrose in strawberry and glucose, fructose and galactose in blueberry). Organically grown strawberry fruits showed, on average, higher levels of sucrose, while leaves from organically produced blueberries accumulated higher levels of fructose and galactose. It seems that sugar metabolism is completely different between strawberry and blueberry, probably due to contrasting plant morphology, and different fruit types.

In general, there are many explanations for why organic/integrated fruits should have higher or lower sugar level in its fruits and leaves. Terry et al. [44] and Bordonaba and Terry [45] claimed that N application and other pre-harvest factors increase sugar level in fruits and leaves. On the other hand, according to Chiou and Bush [46], fruits of organically grown strawberry and blueberry accumulate higher levels of sucrose. As the yields of these two berry fruits are, in most cases, lower for organic production compared to integrated production, the higher levels of sucrose in organically grown berries could be connected with the decreased number of ‘sink’ demands. Conversely, with elevated ‘sink’ demands, like in integrated production, the sucrose level decreases. However, low sink power affects the accumulation of carbohydrates in leaves [47]. Plants that are exposed to abiotic factors, especially stresses (where organic production might be included), could have problems with movement of water through plasma membrane, disrupted osmotic pressure, and depleted transpiration [48]. In this case, plants can develop some physiological and metabolic modifications, such as lower photosynthetic activity, displacement and unusual circulation of organic compounds, as well as the accumulation of primary and secondary metabolite products, especially sugars [49].

#### 2.3.1. Strawberry Fruit

The levels of the two major sugars, glucose and fructose, were somewhat higher in ‘Alba’ and ‘Favette’ from integrated production, in comparison with organic regime. Only in the case of the ‘Clery’, were glucose and fructose more abundant in organically grown berries (57.775 mg/g and 66.238 mg/g, respectively) when compared with integrated production (24.653 mg/g and 46.036 mg/g, respectively). Sucrose, another major sugar, was found in higher concentrations in organic strawberries (up to 68.120 mg/g) than in fruit from integrated production (range from 15.863 mg/g to 38.539 mg/g). Interestingly, in all examined samples, dominant sugars included the monosaccharides glucose or fructose, except organic ‘Favette’, where disaccharide sucrose was the most abundant (68.120 mg/g). Other minor sugar components were mostly found in larger amounts in organic production, especially in organic ‘Clery’. The concentration of monosaccharide galactose, which is almost as sweet as glucose, was up to 2.892 mg/g (in ‘Favette’ org) and was more abundant in two organic strawberries (‘Favette’ and ‘Clery’). Our results are partly consistent with data presented by Cayuela et al. [50] and Conti et al. [51], who found higher sugar content in organic strawberries than in integrally grown fruits, but opposite to those of Hargreaves et al. [52], who found no differences in the sugar content of organically and conventionally grown strawberries.

Other monosaccharides, such as arabinose, ribose, xylose, and rhamnose, were present in lower contents. da Silva et al. [53] showed the presence of rutinose, arabinose, and rhamnose in strawberry fruit as a substituting sugar in strawberry anthocyanins, while Sturm et al. [54] also distinguish xylose, in trace amounts, in some strawberry cultivars.

All three organically grown strawberries stored higher amounts of monosaccharide xylose, but lower levels of arabinose. Lima et al. [55] showed that in heat stressed *Coffea arabica* plants, elevated xylose content (~30%) was accompanied with a reduction in arabinose (~33%), thus demonstrating arabinose swapping with xylose. Also, organic strawberry fruits stored higher levels of turanose (structural isomer of sucrose), which can be increased under the influence of bacterial attack [56]. Additionally, significant differences in isomaltotriose and maltotriose contents were found, but no regularity was found when comparing the type of production.

#### 2.3.2. Strawberry Leaves

In all strawberry leaves, the most abundant sugar was glucose, with a range from 3.696 mg/g (‘Alba’) to 8.047 mg/g (‘Favette’), followed by fructose, in concentrations from 2.242 mg/g (‘Alba’) to 7.065 mg/g (‘Favette’), in all cases favoring the organic regime. The next most prevalent carbohydrate in leaves was sucrose, ranging from 0.904 mg/g (‘Alba’ integrated) to 2.374 mg/g (‘Alba’ organic). All organically produced samples stored higher amounts of sucrose in comparison to the integrally grown one. Also, all the organic leaf samples stored higher amounts of arabinose than integrated leaves. Particularly organic ‘Alba’, which was singled out with a content of arabinose almost ten times higher than the integrated one. On the contrary, all integrated strawberry leaf samples stored higher amounts of trisaccharide panose. Monosaccharide ribose was found in significantly higher amounts only in all three strawberry cultivars. Ribose elevation as a response to stress has been reported previously [57].

#### 2.3.3. Blueberry Fruit

Blueberry fruit was characterized by high contents of glucose (26.415 mg/g–46.495 mg/g) and fructose (22.682 mg/g–43.074 mg/g). With regard to the means of cultivation, these two major sugars were more abundant in organic ‘Bluecrop’ and ‘Nui’, and integrated ‘Duke’. Other monosaccharides, such as ribose, rhamnose, and arabinose, were found in significantly lower concentrations (below 0.149 mg/g) and in almost equal amounts in both regimes.

Maltose, together with trehalose and isomaltotriose, were higher in all organic blueberry fruits. Many scientific studies have proved that maltose is increased in different plants during all kinds of stresses, including high temperatures, osmotic stress, and salt stress [58,59,60].

#### 2.3.4. Blueberry Leaves

Glucose and fructose stood out as the most abundant sugars in blueberry leaves, with average concentrations amounting to 6.710 mg/g and 5.085 mg/g, respectively, followed by galactose and sucrose (ranges: 0.558 mg/g–3.344 mg/g and 0.362 mg/g–1.949 mg/g, respectively). Concentrations of galactose were significantly higher in all leaf samples obtained in the organic regime. The same tendency was observed with some minor components, such as monosaccharides: arabinose and rhamnose, disaccharides: trehalose, melibiose, and gentibiose, trisaccharide isomaltotriose and sugar alcohol galactitol. This was not surprising, since it has already been proved that higher levels of arabinose, rhamnose, galactitol and melibiose are a kind of a reaction of plants to stress [57,61,62]. Stachyose, a tetrasaccharide that was not detected in strawberry leaves, was found in blueberry samples in concentrations from 0.124 mg/g (‘Nui’ organic) to 0.302 mg/g (‘Bluecrop’ organic).

### 2.4. Sweetness Index and Total Sweetness Index

Sugars in strawberry and blueberry fruits are very important for the perception of sweetness and consumer acceptance [44,45]. In this study, strawberry ‘Favette’ from integrated production, ‘Clery’ from organic production, and blueberries ‘Bluecrop’ and ‘Nui’ (both from OP), had significantly higher SI and TSI, which is due to the much higher level of fructose (62.813 mg/g, 66.238 mg/g, 43.047 mg/g and 35.503 mg/g, respectively). Our results were higher than those obtained for strawberry by Crespo et al. [30] and Paparozzia et al. [63], probably due to the different cultivars studied and the adverse microclimate.

### 2.5. Principal Component Analysis

PCA was performed in order to establish possible variability in the sugar profiles of fruit and leaf samples of strawberry and blueberry, with special emphasis on the way of cultivation (organic and integrated). Data of 12 objects (the number of fruit and leaf samples) × variables (quantified carbohydrates) were processed using the covariance matrix with autoscaling.

As for fruit samples, the initial matrix was 12 × 18 and PCA resulted in a five-component model that explained 88.59% of the total variance. The first principal component accounted for 55.91%, the second 11.68%, and the third component 8.24% of the total variance. Strawberry and blueberry formed two distinctive groups along PC1 on the PCA correlation plot (Figure 1A). Higher contents of galactose, arabinose and melibiose, found in blueberry fruits, were the most influential variables responsible for the separation of the berries (Figure 1B). On the other hand, clustering of strawberries was related to higher contents of all the other carbohydrates. Also, some clustering along the PC2 axis could be observed. As for strawberries, integrated ‘Alba’ and both integrated and organic ‘Favette’ stood out on the upper part of the PCA correlation plot. Integrated ‘Alba’ was characterized with the highest amount of ribose when compared with other samples (0.413 mg/g), while ‘Favette’ berries had the highest amount of sorbitol. Organically grown ‘Alba’ and ‘Clery’ were grouped, based on the higher concentrations of maltotriose, isomaltotriose and turanose compared to the rest of the samples. Among blueberries, organic ‘Bluecrop’ and ‘Duke’, together with integrated ‘Duke’, stood out on the upper part of the plot due to their higher amounts of arabinose.

The initial matrix for leaf samples was 12 × 20, and the first six components explained 91.21% of the total variance. The first principal component accounted for 31.99%, the second 22.20%, and the third component 15.34% of the total variance. PCA modeling revealed clustering of blueberry and strawberry leaf samples, along PC1 (Figure 2A). The most important variables responsible for the sample grouping were identified using the loading plots (Figure 2B). Blueberry leaves were characterized by higher contents of panose, ribose, stachyose, galactose, and maltose, while strawberry leaf samples accumulated significantly higher amounts of rhamnose and raffinose. Although a complete differentiation of the leaves according to the type of production was not possible, one cluster could be defined. Namely, organically grown ‘Nui’, ‘Bluecrop’, and ‘Duke’ separated from the integrated fruits along the PC2 axis. Variables with the major influence on this clustering were maltose, galactose, and galactitol. Organic ‘Nui’ and ‘Duke’ had particularly high contents of galactose (3.344 mg/g and 2.192 mg/g, respectively) and maltose (0.394 mg/g and 0.253 mg/g, respectively), compared to all the other investigated leaf samples.

In addition, PCA was performed in order to determine sugars responsible for the separations between leaves and fruits of strawberry and blueberry samples. The six-component model explained 92.15% (for strawberry samples) and 94.97% (for blueberry samples) of the total variance. Factor scores and their loadings were shown in Figure 3 and Figure 4. Both score plots revealed clustering of fruits and leaves into two distinctive clusters along PC1 (Figure 3A and Figure 4A). According to Figure 3B, strawberry leaves separated from fruits based on the presence of carbohydrates exclusively found in leaves, such as galactitol, melibiose, and gentiobiose. Also, higher contents of panose, turanose, and arabinose were characteristic of the strawberry leaves. The loading plot for blueberry samples (Figure 4B) revealed galactose, maltotriose, raffinose, fructose, and glucose to be the main variables responsible for the grouping of blueberry fruits in a separate cluster.

## 3. Materials and Methods

### 3.1. Plant Material

Leaves and fruits from three strawberry (‘Favette’, ‘Alba’ and ‘Clery’) and three blueberry cultivars (‘Bluecrop’, ‘Duke’ and ‘Nui’) were used in this study. Both fruit species were grown in organic (OP) and integrated system (IP) in the village of Pambukovica (44°26’27”N, 19°55’23”E, 157 m altitude), municipality Ub, western Serbia. This region is characterized by a temperate continental climate. Average annual temperature is 11.8 °C and precipitation is 726 mm (571.7 mm during vegetation). The orchards were located very close to each other, and thus represented the same micro-climate zone. Integrated and organic highbush blueberry and strawberry orchards were established on sandy mineral soils that were very uniform in morphological and physical characteristics (color and structure). Organic production was done according to Serbian legislation (fully adjusted to EU standards), while integrated production was done as Integrated Pest Management.

Blueberry orchards were established with the two-year-old nursery bush that was planted in the spring of 2012 at a spacing of 3 m × 1 m. Berry picking was done in 2014 during full ripening. Strawberry orchards were planted in July 2013 (organized on raised double beds covered with black polyethylene folia). Plant spacing was 30 × 30 cm. Fruits were picked fully ripe and analyzed in the first (2014) year after planting.

Farmyard manure (40t/ha every fourth year) and pelleted fertilizer (NPK 3:3:7 + 2% МgO, 60% organic matters and NPK 10:3:3 and 72% organic matters) were used as fertilizers in organic production. In the integrated orchards granulated NPK (14:14:17) and ammonium sulphate (20% N; 24% S) fertilizers, were applied at the beginning of intensive vegetative growth in blueberry. During the spring, blueberry plants were fertilized with NPK fertilizer (20:20:20). Fertigation of the plants with water-soluble NPK fertilizer (11:11:32) was utilized in the cropping stage of vegetation (from beginning of June to the half of July). In strawberry, starter fertilizer (11-44-11, 40 kg/ha) was added during intensive plant growth. During intensive fruit growth and ripening, fertilization was done as previously described in Tomić et al. [32].

In organic production, ‘BoniProtect forte’ (with fungus *Aureobasidium pullulans*) and Serenade MAX (*Bacillus subtilis*), together with some Cu and S preparations, were used as fungicides. Rotenone and azadirachtin was used as an active ingredient of the insecticide in both organic blueberry and strawberry production. In organic blueberry production, hand weeding was applied. In the integrated blueberry orchard, copper compounds and mineral oils were applied at the beginning of the vegetation, captan + pyraclostrobin + boscalid with the addition of insecticide pyrethroide was applied before flowering, and after flowering, pyraclostrobin + boscalid + triazole fungicides and the insecticide neonicotinoid were applied. In strawberry, the first spraying was carried out at the beginning of the vegetation with pesticides based on copper compounds. Before flowering, strobilurins (azoxystrobin), with the addition of insecticide pyrethroide (deltamethrin), were applied against fungus diseases and for broad-spectrum control of chewing and sucking insects. During flowering, control of Botrytis cinerea was performed with a combination of cyprodinil + fludioxonil and fenhexamid.

The trial was set up in a completely randomized design with 3 replications and 5 bushes/plants per replication for each cultivar and in each production system. At harvest, a sample of 20 randomly selected fruits and leaves (from each cultivar/cultivation systems/replication) from all around the bush/plant, were taken and used to analyze sugar profile. The maturity levels were the same for all cultivars. After harvesting, all samples were stored in a freezer at −20 °C until chemical analysis.

### 3.2. Chemicals and Materials

Trehalose, fructose, succrose, maltose, glucose, arabinose, maltotriose isomaltotriose, xylose and were purchased from Tokyo Chemical Industry, TCI, (Europe, Belgium); ribose, raffinose, melibiose, gentiobiose, isomaltose and panose were obtained from Tokyo Chemical Industry, TCI, (Tokyo, Japan); iso-erythritol, galactitol, turanose, galactose, isomaltotriose, mannitol, rhamnose and sorbitol were purchased from Sigma-Aldrich (Steinheim, Germany); sodium hydroxide and sodium acetate trihydrate were from Merck (Darmstad, Germany). All aqueous solutions were prepared using ultrapure water (Thermofisher TKA MicroPure water purification system, 0.055 µS/cm). Standard solutions of glucose, fructose, and sucrose were prepared at 1000 ng/mL concentration, whereas 100 ng/mL was concentration of the rest of compounds. Calibration standards were prepared from the stock solutions by dilution with ultrapure water. The quality-control mixture used for monitoring instrument performance was prepared by diluting standards to concentrations in the range 0.9–100 ng/mL (depending on the concentration in samples).

### 3.3. Preparation of Sample Extracts

The leaves were homogenized and pulverized in an analytical mill (A 11 basic Analytical mill, IKA), and 100 mg of leaf powder was dissolved in 10 mL of ultra-pure water. The solution was treated with ultrasound for 1 h, centrifuged at 10,000 rpm, and supernatant was transferred into vials. Supernatant (1 mL) was diluted 50-fold and kept for further analysis.

The fruit samples (20 berries) were homogenized, and 1 g was weighed and transferred into a 100 mL normal flask and diluted with ultra-pure water to mark. 1 mL of solution was transferred into a normal flask and diluted to 50 mL, filtered through a 0.22 µm syringe filter, and put in vials for further analysis.

### 3.4. Ion Chromatography with Amperometric Detection

DIONEX ICS 3000 ion chromatography system (Dionex, Sunnyvale, CA, USA) consisted of quartenery pump (Dionex, Sunnyvale, CA, USA). Carbo Pac^®^PA100 high-performance anion-exchange column (4 × 250 mm) (Dionex, Sunnyvale, CA, USA) termostated to 30 °C, whereas the composition of mobile phase at the beginning of the analysis was 15% A; from 5 to 5.1 min, the composition changed to 15% A and 2% B; at 12 min, another change was performed; and at 12.1 min, it was 15% A, 4% B, 15% A, 4% B, 81% C; at 20.0–20.1 min, it was 20% A, 20% B 60% C; 20.1–30.0 min 20% A, 20% B 60% C. A, B and C are 600 mM sodium hydroxide, 500 mM sodium acetate and ultrapure water, respectively. The equilibration of the chromatographic system to starting mobile phase composition was 15 minutes before injection of the samples. The injection volume was 25 µL, and it was delivered by the use of autosampler ICS AS-DV 50 (Dionex, Sunnyvale, CA, USA). The gold was the working electrode, while Ag/AgCl was the reference in the pulsed amperometric detector.

To determine the recovery of the method, 25, 50 or 75 ng in 5 mL solutions of fructose, sucrose, glucose, were used to fortify the extracts. Arabinose, rhamnose, ribose, trehalose, maltose, isomaltose, raffinose, melezitoze, panose, mannitol, sorbitol galactitol and erythritol were also added to the extracts, and their concentrations were 2.5, 5.0 and 10 ng in 5 mL. The recoveries of the method were determined using the S/(S_0_ + F) × 100%, where S represents the quantity of sugars or sugar alcohols in the fortified sample, S_0_ in the unfortified sample, and F represents the added amount of the standard.

### 3.5. Sweetness Index and Total Sweetness Index

Sweetness Index (SI) and Total Sweetness Index (TSI) were calculated in order to determine the sweetness perception of fruits. Both approaches are based on the proportion of the individual sugars in fruits, while the difference of the indexes is due to diverse relevance of sugars in sweetness perception [64]. SI was calculated, based on the fact that fructose and sucrose are 2.30 and 1.35 times, respectively, sweeter than glucose, so the equation is:
SI = (1.00 × [glucose]) + (2.30 × [fructose]) + (1.35 × [sucrose]),
(1)

TSI was expressed in such a way that each sugar is estimated relative to sucrose, resulting in the following equation:

TSI = (1.00 × [sucrose]) + (0.76 × [glucose]) + (1.50 × [fructose]),
(2)

### 3.6. Statistical Analysis

The NCSS software package was used to perform statistical analyses (www.ncss.com). Tukey’s test was used to evaluate significant differences (*P* ≤ 0.05) between the mean values. For Principal Component Analysis (PCA), MATLAB PLS Tool Box (Version 7.12.0) was used. Data were group-scaled, and the singular value decomposition algorithm (SVD) and a 0.95 confidence level for Q and Hotelling T2 limits for outliers were chosen.

## 4. Conclusions

To the best of our knowledge, this is the first study that covered comprehensive analysis and comparison of sugar profile and source/sink relationship of fruits and leaves in several cultivars of two berry fruits grown under different production systems.

Sugar metabolism in strawberry and blueberry was quite different, which was proved by PCA, which made a clear separation between those two berry fruit species. None of the ‘main’ sugar components helped us discriminate fruits of strawberry from blueberry, but it was done by galactose, arabinose and melibiose. For leaf recognition between those two species, the most influential were panose, ribose, stachyose, galactose, maltose, rhamnose and raffinose.

The source-sink relationship in strawberry (a non-climacteric fruit) revealed that fruits stored much higher levels of glucose, fructose and sucrose (~12, ~22, and ~66 folds, respectively) than leaves, while in blueberry (climacteric fruit), its fruits accumulated ~6.5-fold more glucose, ~9-fold more fructose and ~12-fold more galactose than leaves. The biggest differences between strawberry fruits and leaves were observed for galactitol, melibiose, and gentiobiose, which were stored only in fruits. In blueberry galactose, maltotriose, raffinose, fructose, and glucose were responsible for the separation of blueberry fruits and leaves in two clusters.

PCA could not separate organic from integrated strawberry production (for both fruit and leaf). Generally, the stress-specific responses of the studied plants were very variable due to the different sensitivity levels and defense strategies of cultivars, which directly influenced the fruit quality.

From a practical point of view, the strawberry cultivar ‘Clery’ and the blueberry cultivars ‘Bluecrop’ and ‘Nui’ gave the best results with respect to sugar analysis in organic production and can be recommended for organic way of growing in temperate growing conditions. This is crucial, since choice of cultivars is one of the most important factors for organic fruit production, for the reason that choosing the wrong one could lead to many problems.

## Figures and Tables

**Figure 1 plants-08-00205-f001:**
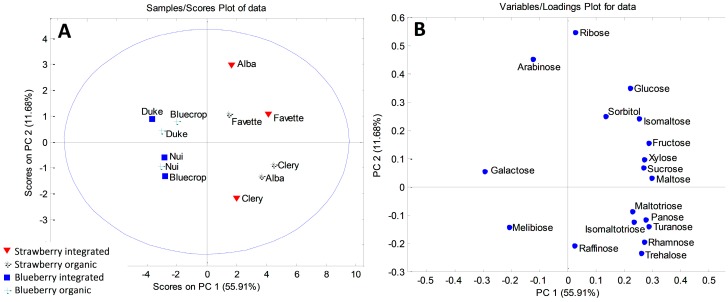
PCA model for sugar-based classification and differentiation of blueberry and strawberry fruits: (**A**) score plot; (**B**) loading plot.

**Figure 2 plants-08-00205-f002:**
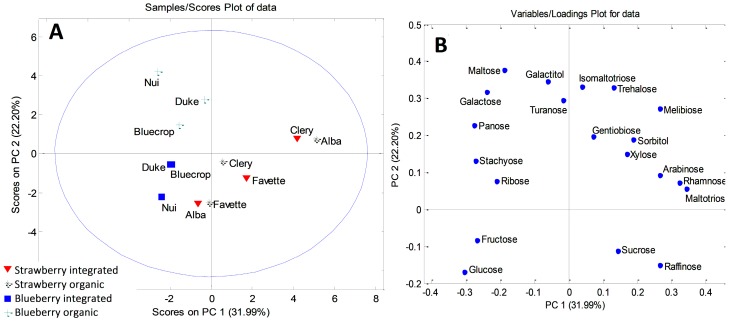
PCA model for sugar-based classification and differentiation of blueberry and strawberry leaves: (**A**) score plot; (**B**) loading plot.

**Figure 3 plants-08-00205-f003:**
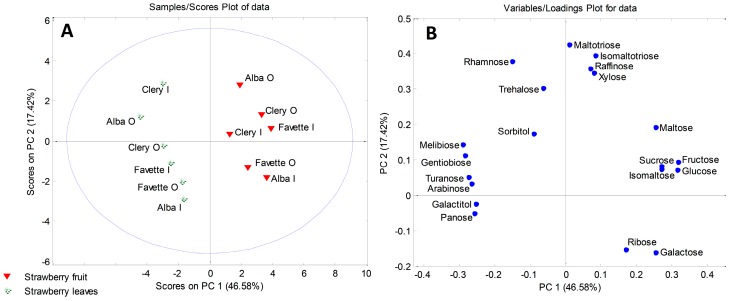
PCA model for sugar-based classification and differentiation of strawberry fruits and leaves: (**A**) score plot; (**B**) loading plot. Organic (O) and integrated (I) production.

**Figure 4 plants-08-00205-f004:**
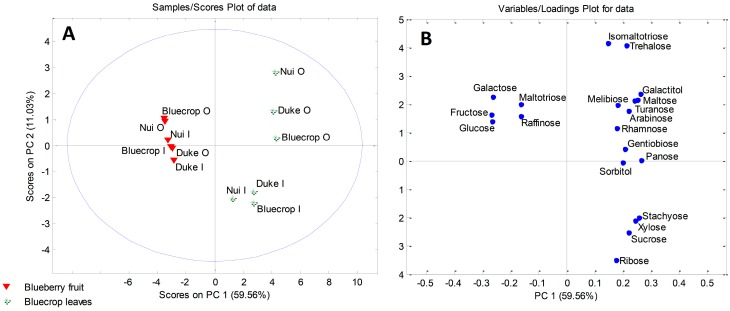
PCA model for sugar-based classification and differentiation of blueberry fruits and leaves: (**A**) score plot; (**B**) loading plot. Organic (O) and integrated (I) production.

**Table 1 plants-08-00205-t001:** Average contents of carbohydrates (mg/g), sweetness index (SI) and total sweetness index (TSI) in integrated (integ) and organic (org) strawberry fruits from three replications.

Carbohydrates		Alba		Favette		Clery	
		integ	org	integ	org	integ	org
1	Glucose	51.975 ± 0.965 ^c 1^	44.288 ± 1.027 ^b^	60.855 ± 1.174 ^d^	46.293 ± 1.117 ^b^	24.653 ± 0.665 ^a^	57.775 ± 0.785 ^d^
2	Fructose	53.534 ± 1.037 ^b^	49.016 ± 0.943 ^ab^	62.813 ± 1.057 ^c^	46.554 ± 0.984 ^a^	46.036 ± 0.885 ^a^	66.238 ± 0.997 ^d^
3	Sucrose	31.715 ± 0.127 ^b^	15.863 ± 0.798 ^a^	68.120 ± 1.855 ^e^	18.285 ± 0.912 ^a^	44.634 ± 0.486 ^d^	38.539 ± 0.832 ^c^
4	Sorbitol	0.014 ± 0.003 ^ab^	0.022 ± 0.005 ^c^	0.061 ± 0.011 ^e^	0.054 ± 0.015 ^d^	0.017 ± 0.001 ^bc^	0.011 ± 0.001 ^a^
5	Trehalose	0.088 ± 0.007 ^a^	0.365 ± 0.023 ^e^	0.243 ± 0.009 ^cd^	0.114 ± 0.009 ^b^	0.297 ± 0.011 ^d^	0.180 ± 0.008 ^c^
6	Arabinose	0.065 ± 0.003 ^d^	0.044 ± 0.002 ^b^	0.057 ± 0.005 ^c^	0.026 ± 0.003 ^a^	0.026 ± 0.002 ^a^	0.021 ± 0.001 ^a^
7	Turanose	0.039 ± 0.002 ^a^	0.129 ± 0.006 ^e^	0.059 ± 0.003 ^b^	0.086 ± 0.005 ^d^	0.071 ± 0.003 ^c^	0.128 ± 0.012 ^e^
8	Galactose	2.396 ± 0.023 ^d^	1.615 ± 0.001 ^c^	1.578 ± 0.011 ^c^	2.892 ± 0.014 ^e^	0.700 ± 0.007 ^a^	1.138 ± 0.012 ^b^
9	Ribose	0.413 ± 0.015 ^e^	0.079 ± 0.017 ^b^	0.101 ± 0.011 ^c^	0.134 ± 0.012 ^d^	0.035 ± 0.005 ^a^	0.036 ± 0.004 ^a^
10	Isomaltose	0.115 ± 0.009 ^c^	0.063 ± 0.001 ^b^	0.251 ± 0.013 ^e^	0.073 ± 0.008 ^b^	0.015 ± 0.002 ^a^	0.148 ± 0.011 ^d^
11	Isomaltotriose	0.034 ± 0.005 ^a^	0.052 ± 0.005 ^b^	0.034 ± 0.003 ^a^	0.042 ± 0.003 ^ab^	0.047 ± 0.005 ^ab^	0.065 ± 0.005 ^c^
12	Maltose	0.267 ± 0.011 ^b^	0.284 ± 0.013 ^b^	0.449 ± 0.025 ^c^	0.194 ± 0.011 ^a^	0.211 ± 0.011 ^a^	0.496 ± 0.010 ^c^
13	Maltotriose	0.102 ± 0.004 ^c^	0.194 ± 0.005 ^e^	0.088 ± 0.002 ^b^	0.097 ± 0.002 ^bc^	0.046 ± 0.002 ^a^	0.158 ± 0.002 ^d^
14	Xylose	0.031 ± 0.001 ^c^	0.038 ± 0.001 ^d^	0.017 ± 0.001 ^a^	0.025 ± 0.001 ^b^	0.022 ± 0.001 ^b^	0.029 ± 0.002 ^c^
15	Panose	0.022 ± 0.001 ^a^	0.021 ± 0.001 ^a^	0.021 ± 0.001 ^a^	0.020 ± 0.001 ^a^	0.026 ± 0.001 ^b^	0.027 ± 0.001 ^b^
16	Rhamnose	0.087 ± 0.002 ^a^	0.268 ± 0.004 ^f^	0.186 ± 0.003 ^d^	0.125 ± 0.003 ^b^	0.234 ± 0.005 ^e^	0.140 ± 0.005 ^c^
17	Raffinose	0.082 ± 0.005 ^a^	0.135 ± 0.005 ^c^	0.121 ± 0.004 ^c^	0.083 ± 0.002 ^a^	0.112 ± 0.002 ^b^	0.108 ± 0.003 ^b^
	Total sugars	140.984 ^b^	112.476 ^a^	195.023 ^d^	115.128 ^a^	117.182 ^a^	165.231 ^c^
	SI	217.919 ^c^	178.440 ^a^	297.287 ^e^	178.052 ^a^	190.792 ^b^	262.150 ^d^
	TSI	151.517 ^c^	123.046 ^a^	208.589 ^e^	123.299 ^a^	132.424 ^b^	181.805 ^d^

^1^ Different letter in the same row denotes a significant difference among cultivars/cultivation systems according to Tukey’s test, *p* < 0.05.

**Table 2 plants-08-00205-t002:** Average contents of carbohydrates (mg/g), sweetness index (SI) and total sweetness index (TSI) in integrated (integ) and organic (org) blueberry fruits from three replications.

Carbohydrates		Bluecrop		Duke		Nui	
		integ	org	integ	org	integ	org
1	Glucose	26.036 ± 1.057 ^a 1^	46.495 ± 1.532 ^c^	35.512 ± 1.117 ^b^	26.415 ± 1.121 ^a^	28.060 ± 1.119 ^a^	34.443 ± 1.742 ^b^
2	Fructose	22.682 ± 1.032 ^a^	43.047 ± 1.011 ^e^	30.920 ±1.285 ^c^	30.516 ±1.117 ^c^	26.420 ± 1.123 ^b^	35.503 ±1.127 ^d^
3	Galactose	7.112 ± 0.214 ^a^	8.727 ± 0.226 ^d^	7.968 ± 0.176 ^bc^	10.112 ± 0.032 ^e^	7.845 ± 0.0216 ^bc^	8.363 ± 0.0313 ^cd^
4	Ribose	0.050 ± 0.003 ^a^	0.075 ± 0.005 ^b^	0.149 ± 0.007 ^d^	0.059 ± 0.004 ^a^	0.080 ± 0.006 ^bc^	0.088 ± 0.006 ^c^
5	Sucrose	0.062 ± 0.003 ^a^	0.067 ± 0.003 ^a^	0.082 ± 0.004 ^c^	0.076 ± 0.003 ^bc^	0.061 ± 0.002 ^a^	0.085 ± 0.003 ^c^
6	Trehalose	0.006 ± 0.001 ^a^	0.028 ± 0.002 ^bc^	0.034 ± 0.002 ^c^	0.037 ± 0.003 ^c^	0.023 ± 0.001 ^b^	0.063 ± 0.002 ^d^
7	Maltose	0.096 ± 0.004 ^c^	0.100 ± 0.003 ^c^	0.010 ± 0.001 ^a^	0.051 ± 0.002 ^b^	0.048 ± 0.002 ^b^	0.062 ± 0.002 ^b^
8	Maltotriose	0.087 ± 0.001 ^c^	0.055 ± 0.001 ^b^	0.031 ± 0.001 ^a^	0.055 ± 0.002 ^b^	0.062 ± 0.002 ^b^	0.050 ± 0.002 ^b^
9	Rhamnose	0.035 ± 0.001 ^c^	0.019 ± 0.001 ^ab^	0.031 ± 0.001 ^c^	0.035 ± 0.001 ^c^	0.013 ± 0.001 ^a^	0.022 ± 0.001 ^b^
10	Raffinose	0.025 ± 0.001 ^b^	0.086 ± 0.001 ^c^	0.046 ± 0.001 ^c^	0.012 ± 0.001 ^a^	0.182 ± 0.001 ^d^	0.264 ± 0.002 ^e^
11	Arabinose	0.016 ± 0.001 ^a^	0.054 ± 0.002 ^cd^	0.067 ± 0.002 ^d^	0.061 ± 0.002 ^cd^	0.042 ± 0.002 ^b^	0.048 ± 0.002 ^bc^
12	Isomaltotriose	0.012 ± 0.001 ^a^	0.049 ± 0.002 ^c^	0.016 ± 0.001 ^a^	0.018 ± 0.001 ^a^	0.028 ± 0.001 ^b^	0.031 ± 0.001 ^b^
13	Melibiose	0.029 ± 0.001 ^b^	0.003 ± 0.001 ^a^	0.031 ± 0.001 ^b^	0.003 ± 0.001 ^a^	0.005 ± 0.001 ^a^	0.042 ± 0.001 ^c^
14	Panose	0.018 ± 0.001 ^b^	0.008 ± 0.001 ^a^	0.008 ± 0.001 ^a^	0.008 ± 0.001 ^a^	0.010 ± 0.001 ^a^	0.009 ± 0.001 ^a^
15	Sorbitol	0.015 ± 0.002 ^a^	0.010 ± 0.001 ^a^	0.014 ± 0.001 ^a^	0.031 ± 0.002 ^c^	0.021 ± 0.001 ^b^	0.004 ± 0.001 ^a^
	Total sugars	56.281 ^a^	98.823 ^c^	74.919 ^b^	67.489 ^ab^	62.900 ^ab^	79.077 ^b^
	SI	78.288 ^a^	145.594 ^c^	106.739 ^b^	96.704 ^ab^	88.908 ^a^	116.215 ^b^
	TSI	53.872 ^a^	99.974 ^c^	73.451 ^ab^	65.925 ^ab^	61.017 ^ab^	79.516 ^b^

^1^ Different letter in the same row denotes a significant difference among cultivars/cultivation systems according to Tukey’s test, *p* < 0.05.

**Table 3 plants-08-00205-t003:** Average contents of carbohydrates (mg/g) in integrated (integ) and organic (org) strawberry leaves from three replications.

Carbohydrates		Alba		Favette		Clery	
		integ	org	integ	org	integ	org
1	Glucose	7.403 ± 0.035 ^c 1^	3.696 ± 0.018 ^a^	5.198 ± 0.024 ^b^	8.047 ± 0.008 ^c^	4.222 ± 0.008 ^ab^	7.682 ± 0.008 ^c^
2	Fructose	5.571 ± 0.011 ^c^	2.242 ± 0.007 ^a^	4.124 ± 0.004 ^b^	7.065 ± 0.006 ^d^	2.755 ± 0.009 ^a^	6.481 ± 0.003 ^cd^
3	Sucrose	0.904 ± 0.011 ^a^	2.374 ± 0.013 ^d^	1.025 ± 0.014 ^a^	1.923 ± 0.011 ^c^	1.196 ± 0.009 ^ab^	1.408 ± 0.006 ^b^
4	Galactose	0.903 ± 0.011 ^d^	0.179 ± 0.009 ^a^	0.586 ± 0.009 ^c^	1.106 ± 0.008 ^d^	0.315 ± 0.012 ^b^	0.353 ± 0.014 ^b^
5	Turanose	0.219 ± 0.011 ^a^	0.208 ± 0.011 ^a^	0.203 ± 0.012 ^a^	0.265 ± 0.0011 ^b^	0.341 ± 0.012 ^c^	0.378 ± 0.014 ^c^
6	Rhamnose	0.104 ± 0.0002 ^a^	0.287 ± 0.002 ^d^	0.189 ± 0.002 ^b^	0.207 ± 0.002 ^b^	0.245 ± 0.002 ^c^	0.187 ± 0.002 ^b^
7	Trehalose	0.180 ± 0.002 ^b^	0.300 ± 0.003 ^c^	0.266 ± 0.003 ^c^	0.068 ± 0.001 ^a^	0.172 ± 0.002 ^b^	0.286 ± 0.002 ^c^
8	Maltose	0.122 ± 0.002 ^a^	0.111 ± 0.005 ^a^	0.107 ± 0.004 ^a^	0.143 ± 0.0008 ^ab^	0.179 ± 0.003 ^bc^	0.196 ± 0.005 ^c^
9	Raffinose	0.083 ± 0.002 ^a^	0.074 ± 0.002 ^a^	0.070 ± 0.002 ^a^	0.08 ± 0.004 ^a^	0.124 ± 0.003 ^b^	0.129 ± 0.002 ^b^
10	Sorbitol	0.035 ± 0.002 ^b^	0.092 ± 0.004 ^d^	0.014 ± 0.001 ^a^	0.016 ± 0.001 ^a^	0.066 ± 0.002 ^c^	0.011 ± 0.001 ^a^
11	Panose	0.054 ± 0.004 ^b^	0.033 ± 0.002 ^a^	0.047 ± 0. 002 ^ab^	0.034 ± 0.002 ^a^	0.056 ± 0.003 ^b^	0.049 ± 0.002 ^ab^
12	Arabinose	0.071 ± 0.004 ^a^	0.677 ± 0.004 ^d^	0.254 ± 0.012 ^b^	0.356 ± 0.016 ^c^	0.226 ± 0.011 ^b^	0.307 ± 0.012 ^bc^
13	Galactitol	0.011 ± 0.002 ^a^	0.039 ± 0.003 ^b^	0.021 ± 0.002 ^ab^	0.016 ± 0.002 ^a^	0.010 ± 0.001 ^a^	0.058 ± 0.003 ^c^
14	Ribose	0.023 ± 0.001 ^a^	0.029 ± 0.002 ^a^	0.025 ± 0.001 ^a^	0.029 ± 0.001 ^a^	0.021 ± 0.001 ^a^	0.095 ± 0.003 ^b^
15	Isomaltotriose	0.025 ± 0.001 ^a^	0.031 ± 0.002 ^a^	0.034 ± 0.001 ^a^	0.031 ± 0.001 ^a^	0.065 ± 0.002 ^b^	0.033 ± 0.001 ^a^
16	Maltotriose	0.026 ± 0.001 ^a^	0.128 ± 0.002 ^c^	0.129 ± 0.002 ^c^	0.045 ± 0.001 ^ab^	0.214 ± 0.002 ^d^	0.057 ± 0.001 ^b^
17	Xylose	0.006 ± 0.001 ^a^	0.028 ± 0.001 ^b^	0.010 ± 0.001 ^a^	0.012 ± 0.001 ^a^	0.035 ± 0.001 ^b^	0.021 ± 0.001 ^b^
18	Melibiose	0.019 ± 0.001 ^a^	0.098 ± 0.001 ^c^	0.064 ± 0.004 ^b^	0.022 ± 0.002 ^a^	0.089 ± 0.003 ^c^	0.054 ± 0.004 ^b^
19	Gentiobiose	0.013 ± 0.001 ^a^	0.045 ± 0.001 ^c^	0.013 ± 0.001 ^a^	0.014 ± 0.001 ^a^	0.030 ± 0.002 ^b^	0.015 ± 0.001 ^a^
	Total sugars	15.772 ^b^	10.671 ^a^	12.379 ^ab^	19.487 ^c^	10.361 ^a^	17.800 ^bc^

^1^ Different letter in the same row denotes a significant difference among cultivars/cultivation systems according to Tukey’s test, *p* < 0.05.

**Table 4 plants-08-00205-t004:** Average content of carbohydrates (mg/g) in integrated (integ) and organic (org) blueberry leaves from three replications.

Carbohydrates		Bluecrop		Duke		Nui	
		conv	org	conv	org	conv	org
1	Glucose	7.009 ± 0.086 ^b 1^	7.226 ± 0.042 ^c^	7.326 ± 0.036 ^c^	4.782 ± 0.022 ^a^	6.922 ± 0.012 ^b^	6.996 ± 0.014 ^b^
2	Fructose	5.621 ± 0.042 ^c^	4.663 ± 0.022 ^b^	5.005 ± 0.022 ^bc^	4.746 ± 0.028 ^b^	4.412 ± 0.018 ^a^	6.065 ± 0.019 ^d^
3	Sucrose	1.186 ± 0.011 ^c^	0.855 ± 0.010 ^b^	1.949 ± 0.011 ^d^	0.362 ± 0.018 ^a^	1.074 ± 0.003 ^bc^	1.191 ± 0.002 ^c^
4	Galactose	0.940 ± 0.003 ^b^	1.518 ± 0.003 ^c^	1.568 ± 0.004 ^c^	2.192 ± 0.004 ^d^	0.558 ± 0.001 ^a^	3.344 ± 0.002 ^e^
5	Ribose	0.413 ± 0.011 ^e^	0.199 ± 0.008 ^cd^	0.282 ± 0.007 ^d^	0.251 ± 0.006 ^d^	0.117 ± 0.002 ^b^	0.069 ± 0.001 ^a^
6	Panose	0.377 ± 0.003 ^a^	0.783 ± 0.004 ^b^	0.684 ± 0.004 ^b^	0.870 ± 0.004 ^c^	0.947 ± 0.003 ^d^	0.891 ± 0.003 ^cd^
7	Turanose	0.252 ± 0.002 ^c^	0.167 ± 0.002 ^b^	0.231 ± 0.002 ^c^	0.274 ± 0.003 ^c^	0.091 ± 0.002 ^a^	0.577 ± 0.002 ^d^
8	Maltose	0.224 ± 0.002 ^b^	0.201 ± 0.002 ^b^	0.186 ± 0.002 ^ab^	0.253 ± 0.003 ^b^	0.141 ± 0.002 ^a^	0.394 ± 0.002 ^c^
9	Stachyose	0.202 ± 0.002 ^c^	0.302 ± 0.003 ^d^	0.174 ± 0.002 ^b^	0.156 ± 0.001 ^b^	0.221 ± 0.002 ^c^	0.124 ± 0.001 ^a^
10	Sorbitol	0.053 ± 0.001 ^c^	0.032 ± 0.001 ^b^	0.026 ± 0.001 ^a^	0.023 ± 0.001 ^a^	0.026 ± 0.001 ^a^	0.058 ± 0.001 ^c^
11	Trehalose	0.066 ± 0.001 ^b^	0.161 ± 0.001 ^d^	0.102 ± 0.001 ^c^	0.361 ± 0.002 ^e^	0.024 ± 0.001 ^a^	0.353 ± 0.002 ^e^
12	Arabinose	0.083 ± 0.001 ^a^	0.378 ± 0.002 ^c^	0.114 ± 0.002 ^ab^	0.201 ± 0.002 ^b^	0.077 ± 0.001 ^a^	0.180 ± 0.002 ^b^
13	Galactitol	0.017 ± 0.001 ^a^	0.041 ± 0.001 ^b^	0.038 ± 0.001 ^b^	0.045 ± 0.001 ^b^	0.011 ± 0.001 ^a^	0.065 ± 0.001 ^c^
14	Isomaltotriose	0.028 ± 0.001 ^a^	0.049 ± 0.001 ^b^	0.031 ± 0.001 ^a^	0.032 ± 0.001 ^a^	0.025 ± 0.001 ^a^	0.070 ± 0.001 ^c^
15	Maltotriose	0.040 ± 0.001 ^c^	0.017 ± 0.001 ^a^	0.030 ± 0.001 ^b^	0.054 ± 0.001 ^d^	0.026 ± 0.001 ^b^	0.041 ± 0.001 ^c^
16	Xylose	0.029 ± 0.001 ^b^	0.014 ± 0.001 ^a^	0.027 ± 0.001 ^b^	0.028 ± 0.001 ^b^	0.012 ± 0.001 ^a^	0.011 ± 0.001 ^a^
17	Melibiose	0.034 ± 0.001 ^b^	0.072 ± 0.002 ^c^	0.026 ± 0.002 ^ab^	0.105 ± 0.002 ^d^	0.015 ± 0.001 ^a^	0.037 ± 0.001 ^b^
18	Gentiobiose	0.022 ± 0.001 ^b^	0.080 ± 0.002 ^c^	0.008 ± 0.001 ^a^	0.028 ± 0.001 ^b^	0.014 ± 0.001 ^a^	0.018 ± 0.001 ^ab^
19	Rhamnose	0.074 ± 0.002 ^c^	0.102 ± 0.002 ^d^	0.086 ± 0.002 ^c^	0.284 ± 0.002 ^e^	0.021 ± 0.001 ^a^	0.046 ± 0.001 ^b^
20	Raffinose	0.023 ± 0.001 ^b^	0.016 ± 0.001 ^ab^	0.026 ± 0.001 ^b^	0.011 ± 0.001 ^a^	0.012 ± 0.001 ^a^	0.022 ± 0.001 ^b^
	Total sugars	16.693 ^ab^	16.876 ^ab^	17.919 ^b^	15.058 ^a^	14.746 ^a^	20.552 ^c^

^1^ Different letter in the same row denotes a significant difference among cultivars/cultivation systems according to Tukey’s test, *p* < 0.05.

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
