# Peer review of "Comparison of Sugar Profile between Leaves and Fruits of Blueberry and Strawberry Cultivars Grown in Organic and Integrated Production System"

_plants, 2019, doi:10.3390/plants8070205_

Reviewer 1 Report

OVERALL

Please check throu the text, since some times the term 'conventional' is used instead of 'integrated': they are not synonimous.

THE FOLLOWING COMMENTS ARE REPORTED IN THE ATTACHED REVISED VERSION:

The UE does not allow to advertise OP produce for better quality, since it is not demonstrated
It must be explained why the two growing systems should be regarded as stressfull or no stressfull
OP production could not be regarded as 'ecological', due to the less yield performance, than forcing to cultivate more land; also, some allowed compunds are not 'ecological': cupper, Rotenone, etc

Author Response

Response to Reviewer #1

We are so thankful for all the comments we got.

We have accepted all the comments and corrected everything that was underlined in pdf version.
We fully understand that we cannot advertise OP produce for better quality, so we deleted that sentence.

Regarding OP as a stressful production we wrote: ‘OP is in most cases considered as a more stressful production system due to insufficient supply of mineral nitrogen, limited number of allowed crop protection products or inefficient application of pesticides which leads to a higher accumulation of primary and secondary metabolic products (Winter and Davis, 2006).’

We fully accept the comment made by the Reviewer #1 that OP production could not be regarded as 'ecological', due to the less yield performance, and usage of cupper and Rotenone, so we deleted this word ‘ecologic’.

Reviewer 2 Report

Both the abstract and the introduction should indicate exactly what problem is to be solved with this comparison. That is, for what this study is carried out. What is to be achieved?. If you want to know the relationship between the sugars of fruits and leaves of the same species and variety, you should make a PCA between the fruits and the leaves. The discussion is too long and the conclusions are not clear.

Author Response

Response to Reviewer #2

We are thankful for all the comment you made, since it made our manuscript better and more informative.

We corrected both ‘Abstract’ and ‘Introduction’ to be more informative and explain what hypothesis and problems we tried to develop and solve.

We also did a PCA between the fruits and the leaves within strawberry and blueberry and added new figures and explanation.
We have cut down the discussion and made conclusions more clear.

Reviewer 3 Report

Revision of “Comparison of sugar profile between leaves and fruits of blueberry and strawberry cultivars grown in organic and conventional system” by Aksic et al.

Title describes precisely the work shown in this manuscript. This analysis is very interesting and could provide solid data to find out if crops cultivated under organic practice possess better food characteristics than those cultivated using conventional systems. The design of the experiment and the acquisition of data seems to have been carefully planned, however I miss the standard deviations in all the data presented in the different Tables. I suppose that the different values represent the mean of different determinations, but the standard deviations of the means and the number of the biological replica used in these calculations should be indicated in the Tables.

Nevertheless, I have another concern that affects to all the results presented in this work. The samples were frozen after harvesting and then homogenized and pulverized in analytical mill and dissolved in water. I think that in this condition multiple enzymatical activities able to act over the different sugars are still operating in the crude extracts. The action of those enzymes could change the levels of the different sugars present in the samples and deeply modified the values determined. I think that another method of extraction should be used (e.g acidic extraction in perchloric acid) to make sure that the levels of sugars are not modified by different enzymatical activities.

Author Response

Response to Reviewer #3

We are thankful for all the comments you gave. It made our manuscript improved.

We have added standard deviations of the means and the number of the biological replica used in the tables.

Regarding the comments about the preparation of samples for the chemical analysis, we can say the following:

      The enzymatic activity of the raw samples and its water extracts is always one of the most important points in sample preparation, but also there is a lot of procedures to handle with it. One of them suggest inactivation of enzymes by the use of acid (such as you suggested), but unfortunately that procedures is inadequate for us because we used highly basic mobile phase and addition of acid could significantly differ the chromatographic condition. The other procedure suggest using methanol-water (7/3 v/v) and evaporation of the supernatant. And of course determination of the samples recovery can also clarify whether the sugars samples content is changed by the enzyme activity. In this study we use the recovery test and in all investigated instances the recovery value was in acceptable range i.e. between 95-105.

Round  2

Reviewer 2 Report

The authors have improved the work and have made the modifications suggested by me.

Reviewer 3 Report

the different points have been addressed